# Changes in CoQ_10_/Lipids Ratio, Oxidative Stress, and Coenzyme Q_10_ during First-Line Cisplatin-Based Chemotherapy in Patients with Metastatic Urothelial Carcinoma (mUC)

**DOI:** 10.3390/ijms232113123

**Published:** 2022-10-28

**Authors:** Patrik Palacka, Jarmila Kucharská, Jana Obertová, Katarína Rejleková, Ján Slopovský, Michal Mego, Daniela Světlovská, Boris Kollárik, Jozef Mardiak, Anna Gvozdjáková

**Affiliations:** 12nd Department of Oncology, Faculty of Medicine, Comenius University, 833 10 Bratislava, Slovakia; 2National Cancer Institute, 833 10 Bratislava, Slovakia; 3Pharmacobiochemical Laboratory of the 3rd Department of Internal Medicine, Faculty of Medicine, Comenius University, 813 72 Bratislava, Slovakia; 4Department of Urology, University Hospital in Bratislava, 851 07 Bratislava, Slovakia

**Keywords:** metastatic urothelial carcinoma, cisplatin, coenzyme Q_10_/lipids ratio, α-tocopherol, thiobarbituric acid-reactive substances

## Abstract

Oxidative stress plays an important role in cancer pathogenesis, and thiobarbituric acid-reactive substance level (TBARS)—a parameter of lipid peroxidation—has prognostic significance in chemotherapy-naive patients with metastatic urothelial carcinoma (mUC). However, the effect of cisplatin (CDDP)-based chemotherapy on oxidative stress, coenzyme Q_10_, and antioxidants remains unknown. The objective of this prospective study was to determine possible changes in the CoQ_10_ (coenzyme Q_10_)/lipids ratio, antioxidants (α-tocopherol, γ-tocopherol, β-carotene, CoQ_10_), total antioxidant status (TAS), and TBARS in plasma at baseline and during first-line chemotherapy based on CDDP in mUC subjects. In this prospective study, 63 consecutive patients were enrolled. The median age was 66 years (range 39–84), performance status according to the Eastern Cooperative Oncology Group (ECOG) was 2 in 7 subjects (11.1%), and visceral metastases were present in 31 (49.2%) patients. Plasma antioxidants were determined by HPLC and TAS and TBARS spectrophotometrically. After two courses of chemotherapy, we recorded significant enhancements compared to baseline for total cholesterol (*p* < 0.0216), very low-density lipoprotein (VLDL) cholesterol (*p* < 0.002), triacylglycerols (*p* < 0.0083), α-tocopherol (*p* < 0.0044), and coenzyme Q_10-TOTAL_ (*p* < 0.0001). Ratios of CoQ_10_/total cholesterol, CoQ_10_/HDL-cholesterol, and CoQ_10_/LDL-cholesterol increased during chemotherapy vs. baseline (*p* < 0.0048, *p* < 0.0101, *p* < 0.0032, respectively), while plasma TBARS declined (*p* < 0.0004). The stimulation of antioxidants could be part of the defense mechanism during CDDP treatment. The increased index of CoQ_10-TOTAL_/lipids could reflect the effect of CDDP protecting lipoproteins from peroxidation.

## 1. Introduction

Bladder cancer (BC) is the tenth most common cancer in the world, and its incidence is steadily rising worldwide [1]. Almost 90% of all BCs are urothelial carcinomas [2]. Patients with muscle-infiltrating BC (MIBC) account for about 30% of all cases. Despite considerable advances in systemic treatment of MIBC during recent years, cisplatin (CDDP) still remains a key agent [3], while up to 50% of patients fail to respond to CDDP-based chemotherapy [4]. Lymph nodes, lungs, liver, bones, and the peritoneum belong among the most common sites of distant metastases in subjects with metastatic urothelial carcinoma (mUC) [3]. The independent factors for predicting survival in mUC patients, as well as performance status assessment using the Eastern Cooperative Oncology Group (ECOG) and visceral metastasis, were established in the 1990s [5].

Plasma cholesterol and TGs are carried by the lipoproteins synthesized in the liver and intestinal cells. Lipoproteins are classified by their density into high density (HDL), low density (LDL), and very low density (VLDL) [6]. LDL-cholesterol was proved to be lower in patients with cancer vs. healthy subjects with comparable BMI, but low LDL-cholesterol levels per se do not cause cancer, and could result from the effects of the tumor on the macroenvironment [7]. Lipids are involved in carcinogenesis. Cancer cells can survive due to the de novo synthesis of lipids without cholesterol efflux [6].

Cholesterol induces the proliferation of cancer cells associated with the incidence of distant metastases [8]. A high expression of LDL receptors in cancer cells indicates high demand for LDL-cholesterol [9]. The uptake of LDL-cholesterol leads to impaired interferon γ (IFN-γ) production and declined cancer cell apoptosis [8]. VLDL and LDL, but not HDL, enhance the malignancy of some cancer cells. VLDL and LDL promote epithelial–mesenchymal transition (EMT) and cancer cell migration via the/a PIK3/Akt/Slug pathway. VLDLs promote distant metastasis [10]. LDL-cholesterol causes a low expression of adhesion molecules such as cadherin-related family member 3, CD226, Claudin 7, and Ocludin genes [11].

CDDP can enter cells via passive diffusion and interact with DNA forming covalent bonds with purine bases, most often with quinine [12]. CDDP also enters cells with the help of transporter proteins—copper transport protein 1 (OCT-1) and OCT-2. Chlorine bound to cisplatin can dissociate, remaining positively charged. CDDP is able to connect to the outer mitochondrial membrane, passing through it and accumulating in the negatively charged membranes of mitochondria, and binding to mitochondrial DNA (mtDNA) which it damages, causing the apoptosis of cancer cells. Therefore, the accumulation of CDDP in the cell leads to the damage of nuclear DNA (nDNA), mtDNA, and also to the electron transport system. CDDP leads to the activation of NADPH, a reduction in antioxidants, and an enhancement in reactive oxygen species (ROS) [13]. ROS, composed of free radical and non-free radical oxygen intermediates—hydroxyl radical, hydrogen peroxide, singlet oxygen, and superoxide—are of great importance in homeostasis and cell signaling. By modulating structural proteins, enzymes, and transcriptional factors, they affect cell proliferation, apoptosis, and tumorigenesis [14].

Endogenous ROS are produced from the mitochondrial respiratory chain during aerobic respiration. The electron transport chain passes electrons to oxygen and reduces oxygen to H_2_O; however, up to 3% of electrons leak from the complex I and the complex III of the respiratory chain and reduce oxygen to free radicals. The other sites of ROS production via cytochrome P450 are the endoplasmic reticulum [15]. Smoking has been identified as a risk factor for BC [16], and N-nitroso-di-butyl-amine, the main component of tobacco which stimulates intracellular ROS-induced oxidative stress and initiates BC under experimental conditions [17], represents an exogenous source of ROS. Increased ROS is associated with a higher risk for cancer, and their modest level is required for cancer cells to survive. However, ROS accumulation in cancer cells results in apoptosis. Moreover, a number of agents can kill cancer cells through ROS induction [18,19,20].

Cellular redox homeostasis is maintained by an endogenous antioxidant defense system containing enzymes (glutathione peroxidase, catalase, superoxide dismutase), glutathione, and free radical scavengers (coenzyme Q_10_ and lipoic acid). The main role of this system is to promote a reduction in lipid peroxide and hydrogen peroxide, and to eliminate the superoxide, preventing cell oxidative damage [14]. Homeostatic levels of CoQ_10_ support cellular functions and survival, while both CoQ_10_ deficiency and CoQ_10_ surplus enhance ROS levels, resulting in mitochondrial dysfunction and cell death [21]. The ROS-induced cell death threshold is likely to differ between cell types [22]. The objective of this prospective study conducted at National Cancer Institute in Bratislava, Slovakia, was to test the hypothesis that CDDP-based systemic therapy could lead to certain changes in plasma lipids, antioxidants (α-tocopherol, γ-tocopherol, β-carotene, coenzyme Q_10-TOTAL_), ratios of CoQ_10_/lipids, and TBARS measured in patients with mUC before initiation and after two courses of chemotherapy.

## 2. Results

### 2.1. Plasma Lipids, Antioxidants, and Lipid Peroxidation Baseline and during Chemotherapy

When values after two courses of chemotherapy were compared to baseline, the changes were determined. There were significant increases in total cholesterol, VLDL-cholesterol, TGs, α-tocopherol, CoQ_10-TOTAL_, CoQ_10-TOTAL_/total cholesterol, CoQ_10-TOTAL_/HDL-cholesterol, and CoQ_10-TOTAL_/LDL-cholesterol, while TBARS levels declined. Other parameters did not change significantly (Table 1).

### 2.2. Plasma Lipids, Antioxidants, and Oxidative Stress Baseline and during Chemotherapy—A Subgroup Analysis by Performance Status

To identify whether changes in individual parameters are affected by performance status, we divided the population of mUC patients into ECOG 0-1 and ECOG 2 subgroups, and evaluated the differences using repeated-measure analysis of variance.

Except α-tocopherol, no significant changes were determined (see Table 2).

### 2.3. Plasma Lipids, Antioxidants, and Oxidative Stress Baseline and during Chemotherapy—A Subgroup Analysis by Visceral Metastases

In the next step, the study population was split into subgroups with visceral metastasis absent or present, and compared as described above. There was no significant difference in any parameter regarding visceral metastasis status (Table 3).

### 2.4. Plasma Lipids, Antioxidants, and Oxidative Stress Baseline and during Chemotherapy—A Subgroup Analysis by Objective Response

To perform this subgroup analysis, the study population was divided by objective (complete and partial) response (OR) to subgroups with absent and present OR. The changes in any of the explored parameters were not affected by OR in mUC patients treated with first-line CDDP-based combined chemotherapy.

Detailed data are shown in Table 4.

### 2.5. Plasma Lipids, Antioxidants, and Oxidative Stress Baseline and during Chemotherapy—A Subgroup Analysis by Serious AEs

Finally, the study population was split into subgroups in which serious AEs were present or absent. As Table 5 shows, the dynamics of any parameter did not depend on serious AEs.

## 3. Discussion

In this prospective study, we revealed a significant increase in total cholesterol, VLDL-cholesterol, and TGs after two courses of CDDP-based chemotherapy compared to baseline in patients with mUC (Table 1). In general, lipids were enhanced regardless of ECOG performance status, the presence (or absence) of visceral metastases, objective response (present or absent), and serious adverse events (present or absent) (Table 2, Table 3, Table 4 and Table 5).

Determining the effect of CDDP-based chemotherapy on plasma lipids was the objective of studies with germ cell tumor patients in the early 1990s. A study by Boyer et al. [23] revealed a significant elevation in serum cholesterol in subjects with metastatic germ cell tumors treated with CDDP-containing chemotherapy when compared to a control population. At the time of lipid measurement, all their patients were in complete remission. Similarly, a study conducted by Raghavan et al. [24] reported that hypercholesterolemia is one of the potential effects of CDDP-based chemotherapy for testicular cancer. However, Ellis et al. did not demonstrate an elevation in total plasma cholesterol after CDDP chemotherapy in a similar population of patients [25]; however, they hypothesized that alterations in plasma lipids could be the result of an enhanced production of cytokines, including tumor necrosis factor (TNF), and varied according to the extent of the disease [23].

After entering the key words “effect”, “cisplatin”, “lipids”, “serum”, “bladder”, and “cancer” into the Pubmed database, we did not find any paper addressing this issue. This could, therefore, be first study to show the effect of CDDP on plasma lipid levels in patients with mUC. Since no patient in this study took lipid-lowering drugs, or smoked during treatment, and no significant changes in BMI were noticed, we believe that the identified changes in plasma lipid levels resulted from systemic therapy. Moreover, they did not depend on its effectiveness, patient performance status, visceral metastasis, or the presence of serious adverse events.

Our study showed a significant rise in both CoQ_10-TOTAL_ and α-tocopherol after CDDP-based chemotherapy vs. baseline in patients with mUC (Table 1). These changes did not depend on performance status, visceral metastasis, objective response, or serious AEs (Table 2, Table 3, Table 4 and Table 5), except for α-tocopherol plasma levels, when the study population was split by performance status. However, this subgroup analysis must be interpreted with caution, as there were only seven patients with ECOG 2 (Table 2).

Because CoQ_10_ is a component of lipoproteins mainly present in the plasma, where about 75% is associated with LDL and the remaining is localized in blood cells (platelets, erythrocytes, and leucocytes) [26], we also calculated ratios of CoQ_10-TOTAL_ and lipids. CoQ_10-TOTAL_/total cholesterol, CoQ_10-TOTAL_/HDL-cholesterol, and CoQ_10-TOTAL_/LDL-cholesterol indexes significantly increased during chemotherapy compared to baseline. On the other hand, plasma levels of TBARS, a parameter of a lipid peroxidation, significantly declined during chemotherapy when compared to the baseline value (Table 1). None of these changes were influenced by the performance status of the patients, the presence of visceral metastases before chemotherapy initiation, objective response achieved by the systemic treatment, or serious AEs related to CDDP-based chemotherapy (Table 2, Table 3, Table 4 and Table 5).

CoQ_10_ plasma levels were an independent prognostic factor that could be used to estimate the risk for pancreatic carcinoma [27] and melanoma progression [28]. Low plasma CoQ_10_ was significantly associated with an increased risk of lung cancer, particularly among current smokers, and may be related to disease progression [29]. Matrix metalloproteinases 2 (MMP-2) plays a key role in cellular invasion and metastasis. Exogenous CoQ_10_ reduces MMP-2 activity, along with the pro-oxidant capacity of cancer cells in a dose-proportionate manner. Mitochondrial ROS is the mediator of MMP-2 activity [30].

A prospective study by Slopovsky et al. [31] showed that low levels of a marker of lipid peroxidation—TBARS—detected in the plasma of chemotherapy-naive patients with mUC correlated with better progression-free survival (PFS) and overall survival (OS). In our previous study [32], the changes in platelet mitochondrial bioenergetics that are key for cell reprogramming in patients with UC were identified. We hypothesized that increased oxidative stress, decreased oxidative phosphorylation (OXPHOS), and a reduced endogenous CoQ_10_ in platelets could contribute to the reprogramming of mitochondrial OXPHOS towards the activation of glycolysis, impaired mitochondrial function, and increased oxidative stress by initiating reverse electron transport from CoQ_10_ to complex I.

Based on the current study’s results, we assume that there is an interaction between CoQ_10_ and CDDP, as it has the ability to bind many hydrogens [13]. CDDP could be a Q-CYCLE proton donor leading to the stimulation of CoQ_10_ production. An increased concentration of CoQ_10_ can stimulate the transport of electrons from complex I and complex II to complex III, and increase mitochondrial ATP production through OXPHOS. Ubiquinone is reduced to ubiquinol, which is a stronger antioxidant, and lipid peroxidation is reduced. Raised CoQ_10_ concentrations can regenerate α-tocopherol and enhance its level.

To conclude, during first-line CDDP-based chemotherapy in patients with mUC, a significant stimulation of lipid fractions (total cholesterol, VLDL-cholesterol, and TGs) and the production of antioxidants (CoQ_10-TOTAL_ and α-tocopherol) along with lipid peroxidation suppression were evident. The enhancement of cholesterol and TGs is not favorable, but the stimulation of antioxidants could represent a host defense mechanism during CDDP treatment. The increased index of CoQ_10-TOTAL_/lipids could reflect the beneficial effect of CDDP in protecting lipoproteins from peroxidation. These findings contribute new insights into the effects of CDDP in patients with mUC.

## 4. Methods

### 4.1. Inclusion/Exclusion Criteria and Study Design

All subjects enrolled into this study met the following inclusion criteria: age ≥ 18 years, a diagnosis of MIBC or muscle-infiltrating urothelial carcinoma of the upper tract (the renal pelvis or ureter) confirmed histologically or cytologically, measurable disease based on RECIST 1.1 criteria, at least one distant metastasis, no prior chemotherapy for inoperable locally advanced or mUC, an ECOG performance status of 0, 1, or 2, and adequate organ function.

Exclusion criteria included disease suitable for local therapy administered with curative intent, a previous malignancy, other than basal or squamous cell carcinomas of the skin, progressing or requiring active treatment within the past 5 years or undergoing potentially curative therapy, in situ cervical cancer, known psychiatric disorders or substance abuse that could have interfered with cooperation with the requirements of this study, known regular use of any illicit drug or a recent history (within the past year) of drug or alcohol abuse, known history of human immunodeficiency virus (HIV), or active hepatitis B or hepatitis C. Concomitant medication with lipid-lowering drugs or triacylglycerol-lowering agents were further exclusion criteria.

We conducted a prospective, non-randomized, single-center observational study to explore specified outcomes outlined in the Introduction section. This study was approved by the Ethical Committee at the National Cancer Institute, Bratislava, Slovakia (protocol code: UC-SK001). All data were entered by investigators into electronic data files and their accuracy was validated for each patient by an independent investigator.

### 4.2. Characteristics of Patients

A total of 63 consecutive patients who met the eligibility requirements were enrolled into this prospective study conducted at the National Cancer Institute in Bratislava (Slovakia). Median age was 66 years (range 39–84 years), and the majority of subjects were male (N = 50, 79.4%). The primary tumor site was the bladder in 82.5% of cases and the upper urinary tract (renal pelvis or ureter) in 17.5%. All patients had pure urothelial carcinoma, and 90.5% of subjects had an Eastern Cooperative Oncology Group (ECOG) performance status (PS) of 0 or 1, and 9.5% scored 2. At least one visceral metastasis was present in 49.2% of all cases. Baseline median body mass index (BMI) was 29.6 kg/m^2^ (range 20.4–34.5 kg/m^2^) without a significant change at week 6.

All subjects were treated with cisplatin 70 mg/m^2^ intravenously on day 1 with gemcitabine 1000 mg/m^2^ intravenously on days 1 and 8. A new treatment cycle started on day 22. The total number of chemotherapy courses was 6. The effect of therapy was evaluated with RECIST 1.1 criteria. Complete response (CR) was recorded in 19.1% and partial response (PR) in 38.1%. The remaining subjects did not respond to systemic therapy.

At least one serious adverse event (AE) was present in 63.5% of subjects. The following grade 3 serious AEs were recorded: neutropenia (20, 31.8%), anemia (11, 17.5%), alopecia (7, 11.1%), hypercreatinemia (6, 9.5%), thrombocytopenia (2, 3.2%), increased values of aspartate aminotransferase (AST) or alanine aminotransferase (ALT) (2, 3.2%), and fatigue (1, 1.6%). There were also observed grade 4 AEs, specifically, neutropenia (5, 7.9%), febrile neutropenia (4, 6.4%), thrombocytopenia (4, 6.3%), hypercreatinemia (1, 1.6%), increased values of AST/ALT (1, 1.6%), and fatigue (1, 1.6%).

Progression-free survival (PFS) was calculated from day 1 of the first course of chemotherapy until disease progression, last follow-up, or death from any cause. Overall survival (OS) was calculated from day 1 of the first course of chemotherapy until last follow-up or death from any cause. At the median follow-up of 10.3 months (range 0.8–142.9 months), 58 patients (92.1%) had progressed and 58 (92.1%) had died. Detailed characteristics of the study population are shown in Table 6.

### 4.3. Plasma Isolation

Peripheral blood samples (12 mL) were collected from all enrolled participants. Samples were collected in Vacutainer^®^ EDTA Blood Collection Tubes (BD Biosciences, Franklin Lakes, NJ, USA) in the morning on day 0 or day 1 before the first and third doses of chemotherapy. Patient blood samples were centrifuged at 1000× *g* for 10 min at room temperature within 2 h of venipuncture. To avoid cellular contamination, plasma was carefully harvested and centrifuged again at 1000× *g* for 10 min at room temperature. The cell-free plasma samples were aliquoted and then cryopreserved at −80 °C. Each sample was thawed only once, immediately before use, for the detection of selected laboratory parameters in the Pharmacobiochemical Laboratory of the 3rd Department of Internal Medicine, Faculty of Medicine, Comenius University in Bratislava, Slovakia.

### 4.4. Selected Laboratory Parameters as a Subject of Interest

The following laboratory parameters were determined in all enrolled patients before systemic treatment initiation and again after two courses of chemotherapy:Lipids (total cholesterol, HDL-cholesterol, LDL-cholesterol; VLDL-cholesterol, TGs), and atherogenic index of plasma;Antioxidants α-tocopherol, γ-tocopherol, β-carotene, CoQ_10-TOTAL_, and total antioxidant status (TAS);The ratios of CoQ_10-TOTAL_ and lipids (CoQ_10-TOTAL_/total cholesterol, CoQ_10-TOTAL_/HDL-cholesterol, CoQ_10-TOTAL_/LDL-cholesterol, and CoQ_10-TOTAL_/TG) were calculated;A marker of lipid peroxidation: (TBARS).

### 4.5. Measurement of Lipids

Peripheral blood (6 mL) for the determination of lipids was collected into Vacutainer^®^ SST^TM^ II Advance (BD Biosciences, Franklin Lakes, NJ, USA) from mUC patients in the morning on day 0 or day 1 before chemotherapy initiation and, thereafter, on day 0 or day 1 before the third course of the same treatment. The samples were processed immediately. TGs, total cholesterol, and HDL-cholesterol were determined by photometry on the Attelica^®^ chemistry analyzer.

TGs were converted into glycerol and fatty acids by the action of lipase. Glycerol was subsequently converted by glycerol kinase into glycerol-3-phosphate and further by glycerol-3-phosphate oxidase into hydrogen peroxide. A colored complex was formed from hydrogen peroxide, 4-aminophenazone, and 4-chlorophenol due to the catalytic effect of peroxidase. The absorbance of the complex was measured as a reaction with an end point at 505/694 nm.

Cholesterol esters were hydrolyzed by cholesterol esterase to cholesterol and free fatty acids. Cholesterol was converted to cholest-4-en-3-one in the presence of oxygen by the action of cholesterol oxidase to form hydrogen peroxide. A colored complex was formed from hydrogen peroxide, 4-aminophenazone, and phenol due to the catalytic effect of peroxidase. The absorbance of the complex was measured as a reaction with an end point at 505/694 nm.

The test to determine HDL-cholesterol consisted of two different reactions. The first was the elimination of chylomicrons, VLDL-cholesterol, and LDL-cholesterol via cholesterol esterase and cholesterol oxidase. The activity of catalase removed the peroxide produced by the oxidase. The second was a specific measurement of HDL-cholesterol after release by the action of surfactant in the 2 D-HDL reagent. The catalase from step 1 was inhibited by sodium azide in the 2 D-HDL reagent. The intensity of the quinonimine coloration produced in the Trinder reaction was directly proportional to the concentration of total cholesterol measured at 596/694 nm.

VLDL-cholesterol was calculated as TGs/2.2, LDL-cholesterol as total cholesterol minus VLDL-cholesterol and HDL-cholesterol, and the atherogenic index of plasma as log_10_(triglyceride/HDL-cholesterol).

### 4.6. Coenzyme Q_10_ and Antioxidants Measurement

Concentrations of CoQ_10-TOTAL_ (ubiquinone + ubiquinol) and lipophilic vitamins (α-tocopherol, γ-tocopherol, β-carotene) in plasma were determined simultaneously by a modified HPLC method with spectrophotometric detection [33,34]. The oxidation of ubiquinol to ubiquinone was performed with 1,4-benzoquinone before analysis [35]. Plasma samples (500 μL) were extracted by a mixture of hexane/ethanol (5/2 *v*/*v*). The tubes were shaken for 5 min and centrifuged at 1000× *g* for 5 min. The hexane layer was separated and the extraction procedure was repeated with 1 ml of the extracted mixture. Collected organic layers were evaporated under nitrogen at 50 °C. The residues were taken up in 99.9% ethanol and injected into a reverse-phase HPLC column. Elution was performed with methanol/acetonitrile/ethanol (6/2/2 *v*/*v*/*v*). The concentration of CoQ_10-TOTAL_ was detected with a UV detector at 275 nm, tocopherols at 295 nm, and β-carotene at 450 nm, using external standards. Data were collected and processed with a chromatographic station. Concentrations of analyzed substances were calculated in μmol/L.

### 4.7. TAS and TBARS Measurements

TAS in plasma was determined using the Randox Total Antioxidant Status kit with colorimetric detection at 600 nm. Concentrations were calculated in mmol/L. TBARS were estimated in plasma after reaction with thiobarbituric acid (TBA), quantified spectrophotometrically at 532 nm and expressed in μmol/L [36].

### 4.8. Statistical Analysis

Data were summarized by frequency for categorical variables and by median ± standard deviation and range for continuous variables. *p* values for categorical variables were calculated using χ^2^ or Fisher’s exact test and for continuous variables the T-test was used for normally distributed values and the Wilcoxon–Mann–Whitney test was used for non-normally distributed values. The subgroup analyses were accomplished using repeated-measure analysis of variance. The value of statistical significance was set to 0.05. All statistical analyses were performed using NCSS 2022 statistical software, Kaysville, UT, USA [37].

## Figures and Tables

**Table 1 ijms-23-13123-t001:** Plasma lipids, antioxidants, index CoQ_10-TOTAL_/lipids, and oxidative stress during chemotherapy in metastatic urothelial carcinoma patients (mUC). N: number of patients; SD: standard deviation; SEM: standard error mean; HDL: high-density lipoprotein; LDL: low-density lipoprotein; VLDL: very low-density lipoprotein; TGs: triacylglycerols; CoQ_10_: coenzyme Q_10_; TBARS: thiobarbituric acid-reactive substances; * significant.

	Time	N	Mean	Median	SD	SEM	*p*
Lipids							
Total cholesterol	Baseline	63	4.56	4.46	1.11	0.15	
(mmol/L)	After CHT	63	5.14	4.96	1.20	0.15	<0.0216 *
HDL-cholesterol	Baseline	63	1.05	1.00	0.40	0.05	
(mmol/L)	After CHT	63	1.15	1.06	0.41	0.05	0.1057
LDL-cholesterol	Baseline	63	2.90	2.90	0.89	0.12	
(mmol/L)	After CHT	63	3.18	3.12	1.00	0.12	0.1440
VLDL-cholesterol	Baseline	63	0.65	0.56	0.32	0.05	
(mmol/L)	After CHT	63	0.83	0.72	0.43	0.05	<0.002 *
TGs	Baseline	63	1.40	1.22	0.70	0.09	
(mmol/L)	After CHT	63	1.71	1.48	0.77	0.09	<0.0083 *
Atherogenic index of plasma	Baseline	63	4.76	4.54	1.65	0.21	
	After CHT	63	4.92	4.65	1.74	0.21	0.5714
Antioxidants							
α-tocopherol	Baseline	63	25.40	24.94	7.45	0.87	
(µmol/L)	After CHT	63	28.46	27.41	6.28	0.87	<0.0044 *
γ-tocopherol	Baseline	63	1.87	1.70	0.81	0.10	
(µmol/L)	After CHT	63	2.01	1.94	0.83	0.10	0.2744
β-carotene	Baseline	63	0.27	0.19	0.27	0.03	
(µmol/L)	After CHT	63	0.29	0.22	0.24	0.03	0.3665
CoQ_10-TOTAL_	Baseline	63	0.45	0.36	0.43	0.06	
(µmol/L)	After CHT	63	0.60	0.50	0.50	0.06	<0.0001 *
Total antioxidant status	Baseline	25	1.35	1.27	0.24	0.04	
(mmol/L)	After CHT	25	1.25	1.24	0.15	0.04	0.1274
Index CoQ_10-TOTAL_/lipids							
CoQ_10-TOTAL_/total cholesterol	Baseline	63	0.10	0.08	0.12	0.02	
(µmol/L/mmol/L)	After CHT	63	0.12	0.10	0.12	0.02	<0.0048 *
CoQ_10-TOTAL_/HDL-cholesterol	Baseline	63	0.50	0.39	0.62	0.08	
(µmol/L/mmol/L)	After CHT	63	0.59	0.46	0.71	0.08	<0.0101 *
CoQ_10-TOTAL_/LDL-cholesterol	Baseline	63	0.17	0.14	0.21	0.03	
(µmol/L/mmol/L)	After CHT	63	0.21	0.16	0.24	0.03	<0.0032 *
CoQ_10-TOTAL_/VLDL-cholesterol	Baseline	63	0.87	0.68	0.81	0.09	
(µmol/L/mmol/L)	After CHT	63	0.85	0.74	0.69	0.09	0.4807
CoQ_10-TOTAL_/TGs	Baseline	63	0.40	0.32	0.37	0.04	
(µmol/L/mmol/L)	After CHT	63	0.41	0.36	0.33	0.04	0.2818
Lipid peroxidation							
TBARS	Baseline	63	5.96	5.86	1.24	0.15	
(µmol/L)	After CHT	63	5.23	4.92	1.15	0.15	<0.0004 *

**Table 2 ijms-23-13123-t002:** Plasma lipids, antioxidants, index CoQ_10-TOTAL_/lipids, and oxidative stress baseline vs. after two courses of chemotherapy in metastatic urothelial carcinoma (mUC) patients—a subgroup analysis by performance status. N: number of patients; SD: standard deviation; SEM: standard error mean; HDL: high-density lipoprotein; LDL: low-density lipoprotein; VLDL: very low-density lipoprotein; TGs: triacylglycerols; CoQ_10_: coenzyme Q_10_; TBARS: thiobarbituric acid-reactive substances; CHT: chemotherapy; ECOG: Eastern Cooperative Oncology Group; * significant.

	Performance Status	Time	N	Mean	Median	SD	SEM	*p*
Lipids								
Total cholesterol	ECOG 0-1	Baseline	56	4.54	4.43	1.12	0.15	
(mmol/L)		After CHT	56	5.09	4.96	1.18	0.16	
	ECOG 2	Baseline	7	4.77	4.68	1.15	0.42	
		After CHT	7	5.56	5.56	1.33	0.45	<0.4215
HDL-cholesterol	ECOG 0-1	Baseline	56	1.06	1.00	0.41	0.05	
(mmol/L)		After CHT	56	1.16	1.08	0.41	0.05	
	ECOG 2	Baseline	7	0.95	0.80	0.26	0.15	
		After CHT	7	1.03	0.95	0.38	0.15	<0.4307
LDL-cholesterol	ECOG 0-1	Baseline	56	2.86	2.79	0.89	0.12	
(mmol/L)		After CHT	56	3.12	3.08	0.97	0.13	
	ECOG 2	Baseline	7	3.16	3.08	0.95	0.34	
		After CHT	7	3.67	3.34	1.25	0.38	<0.2457
VLDL-cholesterol	ECOG 0-1	Baseline	56	0.65	0.55	0.32	0.04	
(mmol/L)		After CHT	56	0.84	0.73	0.44	0.06	
	ECOG 2	Baseline	7	0.66	0.58	0.30	0.12	
		After CHT	7	0.79	0.65	0.37	0.16	<0.8895
TGs	ECOG 0-1	Baseline	56	1.39	1.22	0.71	0.09	
(mmol/L)		After CHT	56	1.71	1.53	0.78	0.10	
	ECOG 2	Baseline	7	1.45	1.28	0.66	0.27	
		After CHT	7	1.72	1.42	0.81	0.29	<0.9038
Atherogenic index of plasma	ECOG 0-1	Baseline	56	4.71	4.47	1.70	0.22	
		After CHT	56	4.79	4.63	1.58	0.23	
	ECOG 2	Baseline	7	5.16	4.94	1.28	0.63	
		After CHT	7	5.95	5.50	2.66	0.65	<0.2053
Antioxidants								
α-tocopherol	ECOG 0-1	Baseline	56	24.83	24.77	6.98	0.98	
(µmol/L)		After CHT	56	27.94	27.25	5.93	0.82	
	ECOG 2	Baseline	7	29.93	28.92	9.99	2.77	
		After CHT	7	32.58	33.82	7.89	2.33	<0.0384 *
γ-tocopherol	ECOG 0-1	Baseline	56	1.87	1.69	0.84	0.11	
(µmol/L)		After CHT	56	2.03	1.97	0.84	0.11	
	ECOG 2	Baseline	7	1.87	2.09	0.68	0.31	
		After CHT	7	1.82	1.52	0.82	0.32	<0.7259
β-carotene	ECOG 0-1	Baseline	56	0.26	0.19	0.28	0.04	
(µmol/L)		After CHT	56	0.28	0.20	0.24	0.03	
	ECOG 2	Baseline	7	0.33	0.31	0.17	0.10	
		After CHT	7	0.38	0.33	0.18	0.09	<0.3124
CoQ_10-TOTAL_	ECOG 0-1	Baseline	56	0.46	0.37	0.45	0.06	
(µmol/L)		After CHT	56	0.62	0.52	0.53	0.07	
	ECOG 2	Baseline	7	0.39	0.36	0.07	0.16	
		After CHT	7	0.46	0.46	0.08	0.19	<0.5437
Total antioxidant status	ECOG 0-1	Baseline	20	1.31	1.27	0.20	0.05	
(mmol/L)		After CHT	20	1.24	1.25	0.11	0.03	
	ECOG 2	Baseline	5	1.52	1.43	0.36	0.10	
		After CHT	5	1.27	1.18	0.26	0.07	0.1515
Index CoQ_10-TOTAL_/lipids								
CoQ_10-TOTAL_/total cholesterol	ECOG 0-1	Baseline	56	0.11	0.09	0.13	0.02	
(µmol/L/mmol/L)		After CHT	56	0.13	0.10	0.12	0.02	
	ECOG 2	Baseline	7	0.09	0.08	0.04	0.05	
		After CHT	7	0.09	0.07	0.02	0.04	<0.545
CoQ_10-TOTAL_/HDL-cholesterol	ECOG 0-1	Baseline	56	0.50	0.39	0.66	0.08	
(µmol/L/mmol/L)		After CHT	56	0.61	0.46	0.75	0.10	
	ECOG 2	Baseline	7	0.45	0.43	0.17	0.24	
		After CHT	7	0.49	0.48	0.14	0.27	<0.7445
CoQ_10-TOTAL_/LDL-cholesterol	ECOG 0-1	Baseline	56	0.18	0.15	0.22	0.03	
(µmol/L/mmol/L)		After CHT	56	0.22	0.17	0.25	0.03	
	ECOG 2	Baseline	7	0.14	0.12	0.06	0.08	
		After CHT	7	0.14	0.12	0.05	0.09	<0.5069
CoQ_10-TOTAL_/VLDL-cholesterol	ECOG 0-1	Baseline	56	0.89	0.69	0.85	0.11	
(µmol/L/mmol/L)		After CHT	56	0.87	0.75	0.72	0.10	
	ECOG 2	Baseline	7	0.71	0.68	0.38	0.14	
		After CHT	7	0.68	0.74	0.26	0.10	<0.5183
CoQ_10-TOTAL_/TGs	ECOG 0-1	Baseline	56	0.41	0.32	0.38	0.05	
(µmol/L/mmol/L)		After CHT	56	0.42	0.36	0.34	0.04	
	ECOG 2	Baseline	7	0.33	0.31	0.17	0.14	
		After CHT	7	0.31	0.34	0.12	0.12	<0.4581
Lipid peroxidation								
TBARS	ECOG 0-1	Baseline	56	5.90	5.64	1.19	0.17	
(µmol/L)		After CHT	56	5.14	4.84	1.10	0.15	
	ECOG 2	Baseline	7	6.45	6.84	1.63	0.47	
		After CHT	7	5.96	5.94	1.41	0.43	<0.1242

**Table 3 ijms-23-13123-t003:** Plasma lipids, antioxidants, index CoQ_10-TOTAL_/lipids, and oxidative stress baseline vs. after two courses of chemotherapy in metastatic urothelial carcinoma (mUC) patients—a subgroup analysis by visceral metastases. N: number of patients; SD: standard deviation; SEM: standard error mean; HDL: high-density lipoprotein; LDL: low-density lipoprotein; VLDL: very low-density lipoprotein; TGs: triacylglycerols; CoQ_10_: coenzyme Q_10_; TBARS: thiobarbituric acid-reactive substances; CHT: chemotherapy.

	Visceral Metastasis	Time	N	Mean	Median	SD	SEM	*p*
Lipids								
Total cholesterol	Absent	Baseline	32	4.67	4.61	0.95	0.20	
(mmol/L)		After CHT	32	5.35	4.98	1.06	0.21	
	Present	Baseline	31	4.45	4.33	1.27	0.20	
		After CHT	31	4.93	4.70	1.30	0.21	<0.2412
HDL-cholesterol	Absent	Baseline	32	1.09	0.99	0.42	0.07	
(mmol/L)		After CHT	32	1.23	1.09	0.42	0.07	
	Present	Baseline	31	1.00	1.00	0.38	0.07	
		After CHT	31	1.07	1.02	0.38	0.07	<0.2043
LDL-cholesterol	Absent	Baseline	32	2.99	3.08	0.76	0.16	
(mmol/L)		After CHT	32	3.32	3.20	0.83	0.18	
	Present	Baseline	31	2.80	2.74	1.02	0.16	
		After CHT	31	3.03	2.81	1.15	0.18	<0.3055
VLDL-cholesterol	Absent	Baseline	32	0.65	0.58	0.32	0.06	
(mmol/L)		After CHT	32	0.84	0.79	0.34	0.08	
	Present	Baseline	31	0.64	0.55	0.32	0.06	
		After CHT	31	0.83	0.64	0.51	0.08	<0.8707
TGs	Absent	Baseline	32	1.38	1.25	0.70	0.12	
(mmol/L)		After CHT	32	1.75	1.56	0.85	0.14	
	Present	Baseline	31	1.41	1.21	0.70	0.13	
		After CHT	31	1.67	1.44	0.70	0.14	<0.8963
Atherogenic index of plasma	Absent	Baseline	32	4.64	4.77	1.33	0.29	
		After CHT	32	4.75	4.61	1.33	0.31	
	Present	Baseline	31	4.88	4.35	1.94	0.30	
		After CHT	31	5.08	4.65	2.10	0.31	<0.4781
Antioxidants								
α-tocopherol	Absent	Baseline	32	26.01	25.68	6.41	1.32	
(µmol/L)		After CHT	32	28.74	27.99	6.56	1.12	
	Present	Baseline	31	24.78	23.46	8.45	1.34	
		After CHT	31	28.17	27.32	6.06	1.14	<0.5467
γ-tocopherol	Absent	Baseline	32	1.86	1.72	0.88	0.15	
(µmol/L)		After CHT	32	2.04	1.94	0.84	0.15	
	Present	Baseline	31	1.87	1.70	0.75	0.15	
		After CHT	31	1.97	2.00	0.84	0.15	<0.8997
β-carotene	Absent	Baseline	32	0.32	0.23	0.33	0.05	
(µmol/L)		After CHT	32	0.31	0.26	0.20	0.04	
	Present	Baseline	31	0.22	0.18	0.19	0.05	
		After CHT	31	0.27	0.19	0.27	0.04	<0.1993
CoQ_10-TOTAL_	Absent	Baseline	32	0.52	0.37	0.58	0.07	
(µmol/L)		After CHT	32	0.66	0.51	0.67	0.09	
	Present	Baseline	31	0.39	0.36	0.15	0.08	
		After CHT	31	0.54	0.50	0.22	0.09	<0.2824
Total antioxidant status	Absent	Baseline	12	1.46	1.39	0.28	0.07	
(mmol/L)		After CHT	12	1.24	1.25	0.15	0.04	
	Present	Baseline	13	1.26	1.25	0.16	0.06	
		After CHT	13	1.25	1.24	0.15	0.04	<0.1425
Index CoQ_10-TOTAL_/lipids								
CoQ_10-TOTAL_/total cholesterol	Absent	Baseline	32	0.12	0.09	0.17	0.02	
(µmol/L/mmol/L)		After CHT	32	0.13	0.10	0.16	0.02	
	Present	Baseline	31	0.09	0.08	0.03	0.02	
		After CHT	31	0.11	0.11	0.04	0.02	<0.3925
CoQ_10-TOTAL_/HDL-cholesterol	Absent	Baseline	32	0.56	0.40	0.86	0.11	
(µmol/L/mmol/L)		After CHT	32	0.63	0.45	0.97	0.13	
	Present	Baseline	31	0.42	0.36	0.19	0.11	
		After CHT	31	0.55	0.49	0.24	0.13	<0.4963
CoQ_10-TOTAL_/LDL-cholesterol	Absent	Baseline	32	0.20	0.14	0.29	0.04	
(µmol/L/mmol/L)		After CHT	32	0.22	0.16	0.32	0.04	
	Present	Baseline	31	0.14	0.15	0.05	0.04	
		After CHT	31	0.19	0.17	0.09	0.04	<0.4463
CoQ_10-TOTAL_/VLDL-cholesterol	Absent	Baseline	32	1.05	0.71	1.07	0.19	
(µmol/L/mmol/L)		After CHT	32	0.90	0.78	0.88	0.16	
	Present	Baseline	31	0.69	0.66	0.31	0.06	
		After CHT	31	0.80	0.72	0.42	0.08	<0.1981
CoQ_10-TOTAL_/TGs	Absent	Baseline	32	0.49	0.33	0.48	0.06	
(µmol/L/mmol/L)		After CHT	32	0.44	0.36	0.42	0.06	
	Present	Baseline	31	0.31	0.30	0.14	0.06	
		After CHT	31	0.37	0.33	0.19	0.06	<0.1291
Lipid peroxidation								
TBARS	Absent	Baseline	32	5.91	5.75	1.21	0.22	
(µmol/L)		After CHT	32	5.02	4.75	0.93	0.20	
	Present	Baseline	31	6.01	6.16	1.28	0.22	
		After CHT	31	5.45	5.19	1.32	0.20	<0.3485

**Table 4 ijms-23-13123-t004:** Plasma lipids, antioxidants, index CoQ_10-TOTAL_/lipids, and oxidative stress baseline vs. after two courses of chemotherapy in metastatic urothelial carcinoma (mUC) patients—a subgroup analysis by objective response (OR). N: number of patients; SD: standard deviation; SEM: standard error mean; HDL: high-density lipoprotein; LDL: low-density lipoprotein; VLDL: very low-density lipoprotein; TGs: triacylglycerols; CoQ_10_: coenzyme Q_10_; TBARS: thiobarbituric acid-reactive substances; CHT: chemotherapy.

	OR	Time	N	Mean	Median	SD	SEM	*p*
Lipids								
Total cholesterol	Absent	Baseline	27	4.39	4.69	1.14	0.21	
(mmol/L)		After CHT	27	4.93	4.55	1.37	0.23	
	Present	Baseline	36	4.69	4.52	1.09	0.19	
		After CHT	36	5.30	5.09	1.04	0.20	<0.2245
HDL-cholesterol	Absent	Baseline	27	0.99	0.90	0.32	0.08	
(mmol/L)		After CHT	27	1.05	0.96	0.33	0.08	
	Present	Baseline	36	1.09	1.01	0.45	0.07	
		After CHT	36	1.22	1.18	0.45	0.07	<0.1515
LDL-cholesterol	Absent	Baseline	27	2.83	2.74	0.93	0.17	
(mmol/L)		After CHT	27	3.11	2.76	1.17	0.19	
	Present	Baseline	36	2.95	3.05	0.87	0.15	
		After CHT	36	3.23	3.14	0.87	0.17	<0.6071
VLDL-cholesterol	Absent	Baseline	27	0.57	0.53	0.30	0.06	
(mmol/L)		After CHT	27	0.77	0.64	0.34	0.08	
	Present	Baseline	36	0.70	0.62	0.32	0.05	
		After CHT	36	0.88	0.80	0.48	0.07	<0.1565
TGs	Absent	Baseline	27	1.27	1.16	0.67	0.13	
(mmol/L)		After CHT	27	1.65	1.38	0.77	0.15	
	Present	Baseline	36	1.49	1.31	0.71	0.12	
		After CHT	36	1.75	1.65	0.78	0.13	<0.3361
Atherogenic index of plasma	Absent	Baseline	27	4.63	4.29	1.31	0.32	
		After CHT	27	5.03	4.79	1.72	0.34	
	Present	Baseline	36	4.85	4.77	1.88	0.28	
		After CHT	36	4.83	4.60	1.78	0.29	<0.9845
Antioxidants								
α-tocopherol	Absent	Baseline	27	23.69	23.11	7.36	1.42	
(µmol/L)		After CHT	27	27.71	27.17	6.27	1.21	
	Present	Baseline	36	26.68	26.54	7.36	1.42	
		After CHT	36	29.02	29.66	6.31	1.05	0.1523
γ-tocopherol	Absent	Baseline	27	1.68	1.59	0.66	0.15	
(µmol/L)		After CHT	27	1.86	1.80	0.81	0.16	
	Present	Baseline	36	2.01	1.96	0.90	0.13	
		After CHT	36	2.11	2.04	0.85	0.14	<0.1123
β-carotene	Absent	Baseline	27	0.28	0.22	0.20	0.05	
(µmol/L)		After CHT	27	0.28	0.23	0.17	0.05	
	Present	Baseline	36	0.26	0.19	0.31	0.05	
		After CHT	36	0.30	0.20	0.28	0.04	<0.9745
CoQ_10-TOTAL_	Absent	Baseline	27	0.37	0.34	0.15	0.08	
(µmol/L)		After CHT	27	0.49	0.46	0.16	0.10	
	Present	Baseline	36	0.52	0.40	0.54	0.07	
		After CHT	36	0.68	0.54	0.64	0.08	<0.1581
Total antioxidant status	Absent	Baseline	14	1.41	1.29	0.28	0.06	
(mmol/L)		After CHT	14	1.25	1.22	0.18	0.04	
	Present	Baseline	11	1.27	1.27	0.16	0.07	
		After CHT	11	1.25	1.27	0.10	0.04	<0.2541
Index CoQ_10-TOTAL_/lipids								
CoQ_10-TOTAL_/total cholesterol	Absent	Baseline	27	0.09	0.08	0.03	0.02	
(µmol/L/mmol/L)		After CHT	27	0.10	0.10	0.03	0.02	
	Present	Baseline	36	0.12	0.09	0.16	0.02	
		After CHT	36	0.13	0.10	0.15	0.02	<0.3130
CoQ_10-TOTAL_/HDL-cholesterol	Absent	Baseline	27	0.40	0.37	0.17	0.12	
(µmol/L/mmol/L)		After CHT	27	0.50	0.47	0.18	0.14	
	Present	Baseline	36	0.56	0.41	0.81	0.10	
		After CHT	36	0.66	0.45	0.92	0.12	<0.3310
CoQ_10-TOTAL_/LDL-cholesterol	Absent	Baseline	27	0.14	0.14	0.05	0.04	
(µmol/L/mmol/L)		After CHT	27	0.18	0.16	0.08	0.05	
	Present	Baseline	36	0.19	0.15	0.28	0.04	
		After CHT	36	0.23	0.18	0.31	0.04	<0.3234
CoQ_10-TOTAL_/VLDL-cholesterol	Absent	Baseline	27	0.82	0.66	0.52	0.10	
(µmol/L/mmol/L)		After CHT	27	0.76	0.69	0.44	0.08	
	Present	Baseline	36	0.91	0.71	0.97	0.16	
		After CHT	36	0.91	0.78	0.83	0.14	<0.4915
CoQ_10-TOTAL_/TGs	Absent	Baseline	27	0.37	0.30	0.24	0.07	
(µmol/L/mmol/L)		After CHT	27	0.36	0.32	0.21	0.06	
	Present	Baseline	36	0.43	0.33	0.44	0.06	
		After CHT	36	0.44	0.37	0.39	0.05	<0.3917
Lipid peroxidation								
TBARS	Absent	Baseline	27	6.12	6.24	1.21	0.24	
(µmol/L)		After CHT	27	5.59	5.60	1.23	0.21	
	Present	Baseline	36	5.84	5.62	1.26	0.21	
		After CHT	36	4.97	4.72	1.03	0.19	<0.1118

**Table 5 ijms-23-13123-t005:** Plasma lipids, antioxidants, index CoQ_10-TOTAL_/lipids, and oxidative stress baseline vs. after two courses of chemotherapy in metastatic urothelial carcinoma (mUC) patients—a subgroup analysis by serious adverse events (AEs). N: number of patients; SD: standard deviation; SEM: standard error mean; HDL: high-density lipoprotein; LDL: low-density lipoprotein; VLDL: very low-density lipoprotein; TGs: triacylglycerols; CoQ_10_: coenzyme Q_10_; TBARS: thiobarbituric acid-reactive substances; CHT: chemotherapy.

	Serious AEs	Time	N	Mean	Median	SD	SEM	*p*
Lipids								
Total cholesterol	Absent	Baseline	23	4.52	4.33	1.28	0.23	
(mmol/L)		After CHT	23	5.12	4.90	1.31	0.25	
	Present	Baseline	40	4.59	4.67	1.02	0.18	
		After CHT	40	5.16	4.98	1.14	0.19	<0.8498
HDL-cholesterol	Absent	Baseline	23	0.95	0.97	0.30	0.08	
(mmol/L)		After CHT	23	1.03	0.99	0.29	0.08	
	Present	Baseline	40	1.11	1.00	0.44	0.06	
		After CHT	40	1.22	1.15	0.45	0.06	<0.0830
LDL-cholesterol	Absent	Baseline	23	2.91	2.90	1.01	0.19	
(mmol/L)		After CHT	23	3.20	3.15	1.04	0.21	
	Present	Baseline	40	2.89	2.88	0.83	0.14	
		After CHT	40	3.17	3.10	1.00	0.16	<0.9210
VLDL-cholesterol	Absent	Baseline	23	0.74	0.75	0.32	0.06	
(mmol/L)		After CHT	40	0.79	0.64	0.50	0.07	
	Present	Baseline	40	0.59	0.55	0.31	0.05	
		After CHT	40	0.79	0.64	0.50	0.07	<0.1097
TGs	Absent	Baseline	23	1.56	1.38	0.72	0.14	
(mmol/L)		After CHT	23	1.93	2.03	0.73	0.16	
	Present	Baseline	40	1.30	1.21	0.68	0.11	
		After CHT	40	1.58	1.38	0.77	0.12	<0.0916
Atherogenic index of plasma	Absent	Baseline	23	5.00	4.74	1.47	0.35	
		After CHT	23	5.19	5.05	1.21	0.36	
	Present	Baseline	40	4.62	4.29	1.75	0.26	
		After CHT	40	4.76	4.32	1.98	0.28	<0.3276
Antioxidants								
α-tocopherol	Absent	Baseline	23	25.05	24.98	6.09	1.56	
(µmol/L)		After CHT	23	28.52	28.45	5.87	1.32	
	Present	Baseline	40	25.60	24.08	8.20	1.19	
		After CHT	40	28.42	27.17	6.57	1.00	<0.8832
γ-tocopherol	Absent	Baseline	23	1.78	1.63	0.93	0.17	
(µmol/L)		After CHT	23	1.94	2.00	0.66	0.18	
	Present	Baseline	40	1.91	1.76	0.75	0.13	
		After CHT	40	2.04	1.86	0.93	0.13	<0.5319
β-carotene	Absent	Baseline	23	0.22	0.19	0.13	0.06	
(µmol/L)		After CHT	23	0.31	0.21	0.31	0.05	
	Present	Baseline	40	0.30	0.20	0.32	0.04	
		After CHT	40	0.28	0.23	0.19	0.04	<0.6576
CoQ_10-TOTAL_	Absent	Baseline	23	0.53	0.34	0.68	0.09	
(µmol/L)		After CHT	23	0.72	0.51	0.78	0.10	
	Present	Baseline	40	0.41	0.37	0.16	0.07	
		After CHT	40	0.53	0.50	0.20	0.08	<0.1960
Total antioxidant status	Absent	Baseline	8	1.27	1.28	0.14	0.09	
(mmol/L)		After CHT	8	1.25	1.28	0.10	0.05	
	Present	Baseline	17	1.39	1.27	0.27	0.06	
		After CHT	17	1.25	1.22	0.16	0.04	<0.4029
Index CoQ_10-TOTAL_/lipids								
CoQ_10-TOTAL_/total cholesterol	Absent	Baseline	23	0.13	0.09	0.20	0.03	
(µmol/L/mmol/L)		After CHT	23	0.15	0.11	0.19	0.02	
	Present	Baseline	40	0.09	0.08	0.04	0.02	
		After CHT	40	0.11	0.10	0.04	0.02	<0.2119
CoQ_10-TOTAL_/HDL-cholesterol	Absent	Baseline	23	0.65	0.41	1.00	0.13	
(µmol/L/mmol/L)		After CHT	23	0.80	0.49	1.13	0.14	
	Present	Baseline	40	0.41	0.38	0.18	0.10	
		After CHT	40	0.48	0.43	0.20	0.11	<0.1025
CoQ_10-TOTAL_/LDL- cholesterol	Absent	Baseline	23	0.21	0.15	0.34	0.04	
(µmol/L/mmol/L)		After CHT	23	0.26	0.18	0.38	0.05	
	Present	Baseline	40	0.15	0.14	0.06	0.03	
		After CHT	40	0.18	0.16	0.09	0.04	<0.2470
CoQ_10-TOTAL_/VLDL-cholesterol	Absent	Baseline	23	0.84	0.57	0.96	0.20	
(µmol/L/mmol/L)		After CHT	23	0.83	0.67	0.96	0.20	
	Present	Baseline	40	0.89	0.72	0.72	0.11	
		After CHT	40	0.86	0.79	0.48	0.08	<0.8556
CoQ_10-TOTAL_/TGs	Absent	Baseline	23	0.40	0.28	0.43	0.08	
(µmol/L/mmol/L)		After CHT	23	0.42	0.32	0.46	0.07	
	Present	Baseline	40	0.40	0.33	0.33	0.06	
		After CHT	40	0.40	0.37	0.22	0.05	<0.9645
Lipid peroxidation								
TBARS	Absent	Baseline	23	6.08	6.16	1.08	0.26	
(µmol/L)		After CHT	23	5.19	5.11	1.11	0.24	
	Present	Baseline	40	5.89	5.62	1.33	0.20	
		After CHT	40	5.26	4.89	1.18	0.18	<0.8431

**Table 6 ijms-23-13123-t006:** Characteristics of study population. N: number of patients; CHT: chemotherapy; AEs: adverse events; GC: gemcitabine + cisplatin; ECOG: Eastern Cooperative Oncology Group; CR: complete response; PR: partial response; * at least one serious AE.

		N	%
Study population		63	100.0
Age (years)	Median (range)		66 (39–84)
Men		50	79.4
Progression		58	92.1
Death		58	92.1
Primary tumor site	Bladder	52	82.5
	Ureter	3	4.8
	Renal pelvis	8	12.7
Histology type	Urothelial carcinoma	63	100.0
Chemotherapy	GC	63	100.0
Performance status	ECOG 0-1	57	90.5
	ECOG 2	7	9.5
Visceral metastasis/es	Present	31	49.2
	Absent	32	50.8
Effect of CHT	CR	12	19.1
	PR	24	38.1
	Stabilization	14	22.2
	Progression	13	20.6
Serious AEs *	Present	40	63.5
	Absent	23	36.5
Progression-free survival (months)	Median (range)		6.0 (0.8–142.9)
Overall survival (months)	Median (range)		10.3 (0.8–142.9)

## Data Availability

The data presented in this study are available on request from the corresponding author.

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
