# Peer review of "Changes in CoQ_10_/Lipids Ratio, Oxidative Stress, and Coenzyme Q_10_ during First-Line Cisplatin-Based Chemotherapy in Patients with Metastatic Urothelial Carcinoma (mUC)"

_ijms, 2022, doi:10.3390/ijms232113123_

Round 1

Reviewer 1 Report

I have reviewed the manuscript titled ""Changes in the CoQ10/lipids ratio, .... metastatic urothelial carcinoma (mU)" authored by Palacka et. al.  The study design was meticulously planned out by the researchers. Before it is accepted for publication, some of the concerns must be addressed.

1. Section 2.1. Characteristics of patients and section 2.2. Selected laboratory parameters as a subject of interest have placed in RESULTS section should be the part of methodology section.

2. The entire manuscript needs to be checked for grammatical, syntax, and language errors. for instance, lines 143, 161, 180, and 197 (repeated use of "and")

3. Although the researchers thoroughly planned and carried out the investigation, they fall short in the writing portion as the Discussion section suffers from a lack of flow while describing or substantiating the results using literature. Authors may have simply merely pasted the lines.

4. The authors reported in lines 325–326 that ethical approval was used for entering the data; what about the ethical approval obtained before this study was carried out? Kindly fix it.

After these minor modifications, the manuscript can, in my opinion, be considered for publication.

Author Response

Dear Editor,

I would like to resubmit our manuscript named “Changes in the CoQ10/lipids ratio, oxidative stress, and coenzyme Q10 during first-line cisplatin-based chemotherapy in patients with metastatic urothelial carcinoma (mUC)” (ijms-1976146) to the journal for a publication. We would like to thank to a reviewer for his/her time and effort. We have followed all suggestions and tried to improve our manuscript accordingly. The revised manuscript (including changes suggested by all reviewers) is attached. Here is our response point by point.  

Q1. Section 2.1. Characteristics of patients and section 2.2. Selected laboratory parameters as a subject of interest have placed in RESULTS section should be the part of methodology section.

A1. Both sections have been replaced.

Q2. The entire manuscript needs to be checked for grammatical, syntax, and language errors. for instance, lines 143, 161, 180, and 197 (repeated use of "and")

A2. The manuscript has been checked for all types of the errors mentioned above.

Q3. Although the researchers thoroughly planned and carried out the investigation, they fall short in the writing portion as the Discussion section suffers from a lack of flow while describing or substantiating the results using literature. Authors may have simply merely pasted the lines.

A3. Both, Introduction and Discussion sections have been revised accordingly.   

Q4. The authors reported in lines 325–326 that ethical approval was used for entering the data; what about the ethical approval obtained before this study was carried out? Kindly fix it.

A4. The ethical approval had been obtained before this study was carried out. This has been stated in the revised 4.1. section clearly (please, see last paragraph).

Note: The other changes of the manuscript were required by second reviewer.  

Reviewer 2 Report

Introduction:

1.       Overall comment: make sure that you only include information in the introduction that is relevant to your study. Too much information can make the reader confused and lose focus.

2.       Line 41: change ‘despite’ to ‘while’

3.       Line 43-45: can you explain briefly why these are of great importance?

4.       Line 46-54: In this paragraph the authors explain the various sources of ROS. This section is somewhat confusing to a person with little knowledge of ROS and could be revised

5.       Line 55: “ROS regulation induces tumorigenesis”. Please soften the language. For example: Increased ROS is associated with higher risk for cancer, or there is a positive association between ROS levels and degree of tumorigenesis.

6.       Line 63-69: The paragraph starts by referring to a manuscript, as such this paragraph should describe the conclusion/findings of this manuscript rather than stating the hypothesis of the manuscript.

7.       Line 70-75: Am I interpreting it correctly that low levels of TBARS = better OS/PFS?

8.       Line 76-79: what is your hypothesis for the study?

Results:

1.       All patients enrolled have metastatic UC. 49.2% have visceral mets. Where do the other 50.8% have metastases?

2.       Line 86-88: Numbers for ECOG scoring does not match information in table 1. Also, in text there is only talk about ECOG score 1, while in the abstract and table it is stated that 9.5% have ECOG score of 2 or higher. The higher than 2 is not reflected in the table, only in the abstract. While in the inclusion/exclusion criteria people are only include with ECOG scores of 0-2.

3.       Line 103-104: “58 patients had progressed and 58 patients died”. Are these the same 58 patients? Did they die of the cancer or other reasons?

4.       Line 109-120: Suggestion: put this information in a table in the methods section

5.       Table 2-6: Suggestion to put all this information in figures/bar graphs. One could have each table be a figure with A-D sub figures for the different categories (lipids, antioxidants etc). The tables could be submitted as supplemental data.

6.       Line 170: the terms ‘absent’ and ‘present’ are confusing here. Maybe call it ‘complete’ and ‘partial’.

Discussion:

1.       There is a fair amount of background information in the discussion. The discussion is intended to discuss what your findings mean, how this relates to previous studies and if the findings support your hypothesis or not. While a short introduction at the beginning of the discussion is appropriate, the majority of the background information should be in the introduction. Try to keep both introduction and discussion concise and to the point.

2.       It is not fully clear if the changes in antioxidant/lipid levels in plasma after 2 rounds of chemo are a result of chemo treatment or if it could mean that the treatment is impacting the cancer. Could you elaborate?

Methods:

1.       Line 310: you should be certain patients meet inclusion/exclusion criteria. Please remove “(or… were required to meet)”.

2.       Plasma isolation: were frozen samples aliquoted and how were they thawed? Were they thawed only once for different experiments or were there multiple freeze-thaw cycles?

3.       Plasma isolation (line 367-369) might want to state that samples were subsequently used for detection of coenzyme Q10, antioxidants etc rather than state that they were ‘processed’.

4.       Statistical analysis: the wrong statistical test appears to be used. The log rank test is used to determine the probability of an event taking place. Here the authors are comparing mean values/concentrations or difference between mean values. Instead, a paired t-test could be used. Also, the way the data is presented now, the authors cannot draw a conclusion on if there is a difference between “absent” and “present” or the different ECOG classifications, which defeats the purpose of splitting the samples up in the different groups. This could be accomplished using a 2-way repeated measures ANOVA.

Author Response

Dear Editor,

I would like to resubmit our manuscript named “Changes in the CoQ10/lipids ratio, oxidative stress, and coenzyme Q10 during first-line cisplatin-based chemotherapy in patients with metastatic urothelial carcinoma (mUC)” (ijms-1976146) to the journal for a publication. We would like to thank to a reviewer for his/her time and effort. We have followed all suggestions and tried to improve our manuscript accordingly. The revised manuscript (including changes suggested by all reviewers) is attached. Here is our response point by point. 

Introduction:

Q1.       Overall comment: make sure that you only include information in the introduction that is relevant to your study. Too much information can make the reader confused and lose focus.

A1. The Introduction section has been revised.

Q2.       Line 41: change ‘despite’ to ‘while’

A2. The suggested change has been made.

Q3.       Line 43-45: can you explain briefly why these are of great importance?

A3. The importance of ROS has been explained in Introduction section (an added brief summary).  

Q4.       Line 46-54: In this paragraph the authors explain the various sources of ROS. This section is somewhat confusing to a person with little knowledge of ROS and could be revised.

A4. The section has been revised.

Q5.       Line 55: “ROS regulation induces tumorigenesis”. Please soften the language. For example: Increased ROS is associated with higher risk for cancer, or there is a positive association between ROS levels and degree of tumorigenesis.

A5. This part of manuscript has been changed as suggested.

Q6.       Line 63-69: The paragraph starts by referring to a manuscript, as such this paragraph should describe the conclusion/findings of this manuscript rather than stating the hypothesis of the manuscript.

A6. This paragraph ahs been modified and included into Discussion section.   

Q7.       Line 70-75: Am I interpreting it correctly that low levels of TBARS = better OS/PFS?

A7. Your interpretation is correct, the manuscript has been clarified in this part (see also A6.).

Q8.       Line 76-79: what is your hypothesis for the study?

A8. The study hypothesis has been clarified.

 Results:

Q1.       All patients enrolled have metastatic UC. 49.2% have visceral mets. Where do the other 50.8% have metastases?

A1. Visceral metastases are considered a prognostic factor for patients with mUC, therefore we stated only the numbers of patient with/without visceral metastases. This is also important for interpretation of this study results. However, if you suggest that percentage of patients with different sites of metastases are relevant for readers, we could add it into the manuscript.    

Q2.       Line 86-88: Numbers for ECOG scoring does not match information in table 1. Also, in text there is only talk about ECOG score 1, while in the abstract and table it is stated that 9.5% have ECOG score of 2 or higher. The higher than 2 is not reflected in the table, only in the abstract. While in the inclusion/exclusion criteria people are only include with ECOG scores of 0-2.

A2. In this study population, there were only 7 patients with ECOG 2. The wrong information (stated only in the abstract) has been corrected.  

Q3.       Line 103-104: “58 patients had progressed and 58 patients died”. Are these the same 58 patients? Did they die of the cancer or other reasons?

A3. In this study population, all patients, who had progressed, died from cancer. For clarification, the definitions of PFS and OS have been added (please, see revised section 4.2. Characteristics of patients).     

Q4.       Line 109-120: Suggestion: put this information in a table in the methods section

A4. This section has been replaced.

Q5.       Table 2-6: Suggestion to put all this information in figures/bar graphs. One could have each table be a figure with A-D sub figures for the different categories (lipids, antioxidants etc). The tables could be submitted as supplemental data.

A5. We believe that revised tables provide better overview than the bar graphs, therefore we would like keep all tables in the manuscript.   

Q6.       Line 170: the terms ‘absent’ and ‘present’ are confusing here. Maybe call it ‘complete’ and ‘partial’.

A6. This part of manuscript has been revised. We believe that revised manuscript is clearer.

Discussion:

Q1.       There is a fair amount of background information in the discussion. The discussion is intended to discuss what your findings mean, how this relates to previous studies and if the findings support your hypothesis or not. While a short introduction at the beginning of the discussion is appropriate, the majority of the background information should be in the introduction. Try to keep both introduction and discussion concise and to the point.

A1. The Introduction and Discussion sections have been revised accordingly.  

Q2.       It is not fully clear if the changes in antioxidant/lipid levels in plasma after 2 rounds of chemo are a result of chemo treatment or if it could mean that the treatment is impacting the cancer. Could you elaborate?

A2. We hypothesize that all described changes resulted from chemotherapy because objective response (present or absent) did not affect the results significantly (please, see also revised Discussion section, 4.8. Statistical analysis section, and Tables 2.–5.)  

Methods:

Q1.       Line 310: you should be certain patients meet inclusion/exclusion criteria. Please remove “(or… were required to meet)”.

A1. The sentence has been changed (please, see revised 4.1 Inclusion/exclusion criteria and study design section). 

Q2.       Plasma isolation: were frozen samples aliquoted and how were they thawed? Were they thawed only once for different experiments or were there multiple freeze-thaw cycles?

A2. This has been explained (clarified) in revised 4.3. Plasma isolation section.  

Q3.       Plasma isolation (line 367-369) might want to state that samples were subsequently used for detection of coenzyme Q10, antioxidants etc rather than state that they were ‘processed’.

A3. This has been modified in revised 4.3. Plasma isolation section.  

Q4.       Statistical analysis: the wrong statistical test appears to be used. The log rank test is used to determine the probability of an event taking place. Here the authors are comparing mean values/concentrations or difference between mean values. Instead, a paired t-test could be used. Also, the way the data is presented now, the authors cannot draw a conclusion on if there is a difference between “absent” and “present” or the different ECOG classifications, which defeats the purpose of splitting the samples up in the different groups. This could be accomplished using a 2-way repeated measures ANOVA.

A4. The subgroup analyses were accomplished using repeated measures analysis of variance. The section 4.8. Statistical analysis, Tables 2.–5., and Discussion section were revised accordingly.  

Round 2

Reviewer 2 Report

No further comments